# Effects of Exercise and Diet on Body Composition and Physical Function in Older Hispanics with Type 2 Diabetes

**DOI:** 10.3390/ijerph18158019

**Published:** 2021-07-29

**Authors:** Edgar Ramos Vieira, Fabricia Azevedo da Costa Cavalcanti, Fernanda Civitella, Monica Hollifield, Stephanie Caceres, Jorge Carreno, Trudy Gaillard, Fatma G. Huffman, Jorge Camilo Mora, Marcos Roberto Queiroga

**Affiliations:** 1Department of Physical Therapy, Florida International University, Miami, FL 33179, USA; Fernanda.civitella@gmail.com (F.C.); scace001@fiu.edu (S.C.); jcarr015@fiu.edu (J.C.); 2Department of Physical Therapy, Federal University of Rio Grande do Norte, Natal 59075-000, RN, Brazil; facnat@yahoo.com; 3Department of Dietetics and Nutrition, Florida International University, Miami, FL 33179, USA; mholl058@fiu.edu (M.H.); huffmanf@fiu.edu (F.G.H.); 4Department of Undergraduate Nursing, Florida International University, Miami, FL 33179, USA; tgaillar@fiu.edu; 5Department of Humanities, Health and Society, Florida International University, Miami, FL 33179, USA; jcmora@fiu.edu; 6Department of Physical Education, Midwestern Parana State University, Guarapuava 85040-167, PR, Brazil; queiroga@unicentro.br

**Keywords:** ethnicity, community dwelling, older adults, diabetes, ethnic groups

## Abstract

Type 2 Diabetes mellitus (DM2) affects 9.3% of the U.S. population. Health disparities are evident in DM2; twice as many Hispanics as non-Hispanic Whites have DM2. The objective of this study was to pilot test the feasibility of implementing and evaluating trends of nutrition and exercise interventions to improve diabetes management and physical function in 29 disadvantaged older Hispanics with DM2. We delivered combined diet and exercise (*n* = 8) and diet-only (*n* = 6) interventions and compared the results to a control/no intervention group (*n* = 15). We cluster-randomized the participants into the three arms based on the senior center they attended. The interventions were delivered twice a week for 3 months (24 sessions) and assessments were conducted pre and post intervention. The results indicate the feasibility of implementing the interventions and slight improvements in both intervention groups compared to the control group. The diet-only group tended to have larger improvements on body composition measures (especially in muscle mass), while the diet + exercise group tended to have larger improvements on physical function (especially in chair stands). There was a high rate of attrition, especially in the diet + exercise group, but those who completed the intervention tended to have improvements in body composition and physical function.

## 1. Introduction

Diabetes, a major health concern in the U.S., affects 9.3% of the general population and disproportionately affects minorities; 1.6 as many Hispanics have diabetes compared to non-Hispanic Whites [1]. The American Diabetes Association and the Association of Diabetes Care & Education Specialists recommend ongoing and patient-centered diabetes self-management education and support for persons with diabetes [2,3]. The standard of care for diabetes is education on proper diet, physical activity, and glucose monitoring [3,4]. However, diabetes education programs are less likely to be available and more likely to close in underserved communities due to lack of referrals and reimbursement [4]. In addition, a prospective study has shown that neighborhood socioeconomic deprivation is associated with decreased physical activity and increased sedentary behavior in older adults [5]. Older adults (≥65 years) with type 2 diabetes follow the same nutrition and physical activity guidelines as the rest of the population [6]. The standards of care may not be sufficient to prevent diabetes complications and the rate of diabetes-related hospitalizations has doubled in the past 20 years from 1273/100,000 in 1993–1995 to 2308/100,000 in 2012–2014 [7].

Type 2 diabetes is associated with physical limitations, gait and balance problems, and falls, especially among older Hispanic adults [8,9]. Falls are the number one cause of injury, hospitalization, and injury-related disability in older adults [10,11]. One in three older adults fall each year and injuries occur in 40–60% of the falls [12]; ~25% of all falls result in emergency department or primary care physician visits (2.4 million cases in the U.S. in 2011, 37.3 million cases/year worldwide) [13,14,15,16]. In the U.S., every 19 min an older adult dies due to a fall and every 11 s an older adult is treated in an emergency department due to a fall-related injury [17]. In Florida, falls are responsible for ~30,000 hospitalizations/year [11]. In a longitudinal study, 68% of fallers had injuries, 24% needed health care, and 35% reported functional decline [18]. Approximately 40% of all nursing home admissions are related to falls [19,20] and ~424,000 people die globally due to falls/year [15,16]. Walking slower than 0.8 m/s and not being able to complete at least 8 chair-rises in 30 s indicates disability and increases the risk of falls and frailty [21]. Older adults with disabilities should undergo comprehensive evaluation and treatment to prevent falls and related injuries [22]. However, Medicare does not cover preventative evaluation and treatments, and falls remain a significant problem both in magnitude and costs.

Falls in older adults can result from impairments in gait, balance, and lower limbs [23]. In more than 25% of cases, falls trigger a cycle of fear of falls, reduced physical activity, deconditioning, functional decline, social isolation, reduced quality of life, depression, and increased risk of subsequent falls (RR = 2.1, 95%CI = 1.3–3.3) [24,25]. Exercise and physical therapy help retrain, recover, and improve balance, strength, and gait, as well as reduce falls and fear of falls in community-dwelling older adults (risk ratio = 0.87, 95% CI = 0.81–0.94) [26]. Programs to reduce falls and fear of falls can decrease medical costs [27,28,29,30]. Exercise-based fall prevention programs for community-dwelling older adults improve physical health, reduce falls and fear of falls, and are cost effective [30,31,32,33,34]. Providing medications and medication management alone (without exercise and diet) did not reduce A1C in uninsured, underserved minorities with diabetes [35]. However, combining nutrition counseling and exercise resulted in a 1% reduction in hemoglobin A1C in underserved middle-aged Hispanics with type 2 diabetes [36]. In addition, an 18-month exercise and nutrition intervention in socioeconomically disadvantaged 48- to 72-year-old rural African Americans with diabetes resulted in a 0.5% reduction in A1C [37]. Increased muscle-mass from exercise and nutrition is associated with lower insulin resistance [38]. However, no nutrition and exercise trials for older Hispanics with type 2 diabetes have been reported, even though several interventions in other populations have shown promise.

The estimated cost of diabetes in 2012 was $245 billion dollars. Diabetes uses 1 in 5 health care dollars and 1 in 3 Medicare dollars; medical expenditures are 2.3 times higher than what they would be in the absence of diabetes [39]. The standard of care for people with diabetes is a medical evaluation once a year with referrals as needed [2,4]. This is not enough, and the rate of hospitalizations due to diabetes has doubled in the past 20 years [7]. Type 2 diabetes is associated with falls [8]. In addition to exercise, nutrition may help in reducing falls. Therefore, the objective of this study was to pilot test the feasibility of implementing and evaluating trends of nutrition and exercise interventions to improve diabetes management and physical function in disadvantaged older Hispanics with type 2 diabetes.

## 2. Methods

### 2.1. Participants

Twenty-nine older Hispanics with type 2 diabetes participated in the study. They were recruited in three senior centers located in disadvantaged neighborhoods. The inclusion criteria were: to be Hispanic; to be ≥65 years old; to have type 2 diabetes; to pass the Mini Cog test [40]; and to score >17 on the Mini Nutritional Assessment [41]. The exclusion criteria were to not be able to exercise due to medical issues or physical limitations; to be wheelchair-bound or on dialysis; to have liver disease, cancer, HIV/AIDS; or to have other physical or psychological conditions preventing participation in a diet or exercise program.

All procedures performed were in accordance with the ethical standards of the institutional and/or national research committee and with the 1964 Helsinki declaration and its later amendments or comparable ethical standards. The study procedures and protocols were approved by the Institutional Review Board (IRB-19–0037). Before the assessment, all questions were addressed, and the participants signed an informed consent form.

### 2.2. Interventions

We cluster-randomized the participants into the 3 arms: diet + exercise (*n* = 8), diet (*n* = 6) and control group (*n* = 15) based on the senior center they attended. The senior centers were in socio-economically disadvantaged neighborhoods and served publicly funded congregate meals. The interventions were delivered at the senior centers twice a week in the afternoons, and attendance was recorded. Hispanic undergraduate and graduate students were trained and helped deliver the exercise (physical therapy students) and nutrition (dietetics and nutrition students) sessions under supervision. The control arm did not receive any intervention as part of the study. A description of the trial arms is presented in Table 1.

The exercise intervention was adapted from the evidence-based Otago Exercise Program for fall prevention in older adults [42]. The exercises were tailored to the needs and capabilities of the participants and involved short conversations (~5 min) about physical function, fall risks and prevention, exercise effects on diabetes, and other exercise-related topics, followed by 30-min exercise sessions.

The educational nutrition intervention was based on Dietary Guidelines for Older Americans and Diabetes.org’s Nutrition Recommendations; it was also tailored to the participants based on food preferences, affordability, and availability. The educational intervention was on healthy eating to improve diabetes management (e.g., glycemic control and diet quality). No meals were provided directly to the participants. The participants received evidenced-based diet education for older adults with type 2 diabetes; the following topics were presented interactively 2 times/week (30 min/session):What’s in food for me? Senior nutrition and food safety;Healthy foods/superfoods/healthy foods on a budget;Meal plans, portion & serving sizes, and reading food labels;Modifying recipes—healthy recipes your way;Dietary supplements and tips for staying hydrated;Closer look at reading food labels & meal planning (tips on eating out).

Participants in the two intervention groups (diet + exercise and diet-only) participated in a one-hour long diabetes education session as part of orientation at the beginning of the program. The interventions (diet + exercise and diet-only) were delivered twice a week for 3 months (24 sessions), and assessments were completed at baseline and 3 months.

### 2.3. Procedures and Data Collection

Data collection took place in the senior centers. All researchers involved in data collection were trained and practiced the standardized data collection and measurement procedures.

#### 2.3.1. Anthropometric and Body Composition

Demographic data were documented; weight, height, and body composition (bioelectric impedance) were measured with the subject wearing light clothing without shoes. Height was measured using a SECA stadiometer (Seca Corp., Columbia, MD, USA) with a precision of 0.1 cm. Body mass index (kg/m^2^) was calculated as weight (kg) divided by height squared (m^2^). Bioelectric impedance analysis was done using the InBody (Biospace, Inc., Los Angeles, CA, USA model 529) to calculate the percentage of muscle mass and body fat (BF). Waist circumference (WC) was measured at the midpoint between the last ribs and the iliac crest and hip circumference (HC) was measured as the maximum circumference around the buttocks posteriorly and the symphysis pubis anteriorly. The circumferences (cm) were measured twice, and the mean was used for analysis. A flexible, non-elastic metric tape measure with a precision of 1 mm was used to take the measures with the individual standing in the anatomical position.

#### 2.3.2. Grip Strength

Grip strength was measured using a digital handheld dynamometer (Jamar^®^) positioned in line with the forearm. The subjects were seated upright in an armless chair with their forearms parallel to the ground and 90° degrees of elbow flexion. The participants were asked to squeeze the handle as hard as possible. Two measurements were taken for each hand with an interval of 1 min between trials. The average of the four measurements was used in the analysis (mean values of right and left hand).

#### 2.3.3. Sit to Stand Test

Participants were seated on a chair without armrests, with their arms crossed across the chest. They were instructed to rise from the chair into standing and sit down again as quickly as possible repeatedly for 30 s without using the arms for assistance. The total number of correct executions (repetitions) was recorded [43].

#### 2.3.4. Blood Pressure

Systolic and diastolic pressure was measured twice in a sitting position. The same trained nurse took both measurements after a 15-min rest using random zero sphygmomanometer (Tycos 5090–02 Welch Allyn Pocket Aneroid Sphygmomanometer, Arden, NC, USA) and a stethoscope (Littmann Cardiology, 3M, St. Paul, MN, USA).

#### 2.3.5. Capillary Glucose and Glycated Hemoglobin Parameters

Blood samples were collected by the same trained nurse and blood glucose and glycated hemoglobin (HbA1c) was assessed with a previously validated capillary glucose test. We used the Seimens Diagnostic-DCA Vantage A1C Analyzer and Reagent Kits (A1C), and the One Touch Verio meter to measure fingerstick glucose.

### 2.4. Statistical Analysis

The Shapiro−Wilk test was utilized to verify data normality and the Levene test for equal variances. The results are expressed in values of mean and standard deviation (±). The comparison between the three groups (diet + exercise, diet-only, and control) was performed from the ANOVA and Tukey’s multiple comparison test HSD. All P-values are two-sided, and the analyses were conducted in a commercial statistical package (SPSS version 25.0—IBM Corporation, Armonk, NY, USA), adopting a significance level of *p* < 0.05.

## 3. Results

Twenty-nine participants completed the first and second assessment in diet + exercise (*n* = 8; 157 ± 9 cm), diet (*n* = 6; 157 ± 15 cm), and control (*n* = 15; 160 ± 8 cm) groups. The comparison (mean ± standard deviation) of body composition measures is presented in Table 2 per assessment (baseline and month 3) and per group. The diet group decreased in the waist-to-hip ratio and increased in muscle mass percentage (3%). The diet + exercise and control group had minimal changes over time.

The upper and lower body strength measures are presented in Table 3, per assessment and per group. The strength indicators tended not to change in the groups over time. The diet + exercise group demonstrated a significantly higher number of chair stands completed in 30 s in month 3.

Diabetes biomarker (HbA1c, random capillary glucose) and blood pressure (systolic and diastolic) data are presented in Table 4, per assessment and per group. HbA1c (%) tended to remain unchanged in all groups and capillary glucose (g/dL) tended to increase in the diet + exercise group and decrease in the diet group.

The preliminary findings indicate some differential effects among the groups and no effects in the control group, as expected. Table 5 presents a summary of the trends observed per variable, per group. Some of these trends were not statistically significant, as indicated on the table. These are the trends presented in detail in the previous tables.

## 4. Discussion

The purpose of this study was to pilot test the feasibility of implementation and the preliminary effectiveness of nutrition and exercise interventions to improve diabetes management and physical function in disadvantaged older Hispanics with type 2 diabetes. Our findings showed that there was no improvement in variables of diabetes management but showed a slight improvement in physical functions from baseline to 3 months. Improvements in physical function can help prevent falls, thus lowering the risk of injuries and frailty [44]. In body composition measures, the diet group decreased in the waist-to-hip ratio (1.02 ± 0.10 vs 0.94 ± 0.12; *p* = 0.048) and increased in muscle mass percentage (26 ± 6 vs. 29 ± 5%; *p* = 0011). For upper and lower body strength measures, the diet + exercise group demonstrated a higher number of chair stands (9 ± 4 vs. 10 ± 5 rep; *p* = 0.025). The control group had minimal changes over time. Considering that the intervention lasted three months (≈24 sessions), these results have clinical significance. It is possible that we could have seen improvements in diabetes management if we were able to continue the intervention period for another three months; unfortunately, we had to stop due to the COVID-19 pandemic. Future studies should evaluate longer intervention periods that may maximize the effects.

Abdominal obesity (visceral fat depots), measured by waist circumference (99 ± 13 cm), was significantly and independently associated with an increase in the prevalence of type 2 diabetes among older Mexican adults [45]. Waist circumference is an indicator associated with visceral fat depots, and visceral fat is more dangerous than subcutaneous fat because visceral fat cells release proteins that contribute to inflammation, atherosclerosis, dyslipidemia, and hypertension [46]. Consequently, visceral adipose tissue may be more closely associated with type 2 diabetes than other indices of obesity [47]. The relationship of muscle mass and fat mass with insulin resistance and metabolic syndrome was evaluated with 14,807 adult participants aged between 18 and 65 in the Korea National Health and Nutrition Examination Survey. The high muscle/low fat was associated with significantly lower insulin resistance [48]. Therefore, the decrease in the waist-to-hip ratio and increase in muscle mass percentage in the diet group is an important indicator of reduced metabolic risk.

Although the diet + exercise group received a broader intervention (nutrition education and physical exercise), except for an increase in performance in the chair stands test (lower body strength indicator), there were no additional gains compared to the diet-only group. Performance tests in the 30-s chair stand test are a cornerstone for detecting early declines in functional independence [49]. In their manuscript, Cesari et al. [50] demonstrated that poor performances on chair stand tests were predictive of the risk of adverse health events in older persons. Type 2 diabetes is associated with falls [8]; it is a significant contributor to increased frailty in older Hispanics (Mexican Americans) [51], and Hispanic women reported the lowest physical activity levels [52]. Therefore, the greater strength of lower limbs can be an important sign of quality of life for the elderly in the diet + exercise group. However, previous education program for Latinos with diabetes did not find significant changes in physical activity, although patient-focused physical activities were introduced [53,54]. Interventions for engaging Hispanics with type 2 diabetes in physical activity may need to focus more on specific action plans that emphasize the time, place, frequency, intensity, and type of exercise, especially for older adults, who may not understand the value of physical activity in the control of their diabetes as well as the potential to prevent falls.

The nutrition and exercise interventions did not improve diabetes biomarkers and blood pressure in the groups (diet and diet + exercise group). The pilot study examined the efficacy of an 8-week culturally tailored intervention focusing on diabetes self-management for Hispanic adults with diabetes and their family members [53]. The findings indicated that the intervention had positive effects on participants’ systolic blood pressure; diabetes knowledge; diabetes self-efficacy; self-management of general diet, specific diet, blood sugar testing, foot care, and fruit and vegetable consumption; as well as both the physical and mental components of health-related quality of life. Clinical improvements were found in participant with diabetes’ HbA1c, waist circumference, and LDL, but these were not statistically significant. In another study, intensive interventions tailored to low-income Latinos showed clinically important short-term improvements in glucose control and glucose variability. However, the authors admit the need for strategies to sustain these improvements [54]. An education program (3 months) resulted in significant increases in both moderate and high-intensity physical activity among a group of Hispanic diabetics and significant reductions in A1C, total cholesterol, LDL, and waist circumference compared to baseline [55].

A limitation of this study was that the small size of the groups reduced the potential to detect significant changes at post-intervention and 3-month follow-up while also limiting the generalizability of the results. In addition, we did not objectively measure food consumption, daily physical activity behaviors, and the patients’ adherence to their medication. However, from the perspectives of patients, their families/caregivers, healthcare professionals, and/or other stakeholders, the facilitators and barriers to successful management of type 2 diabetes are highly dependent on factors that are beyond the control of the individual patients. Successful control requires emphasis on public policies to reinforce health care access, resources, and the promotion of patient-centered care, as well as a health-promoting infrastructure and physical environmental [56]. Adherence to diabetes self-care is poor among Hispanic Americans. Hispanic American participants of a tailored telemedicine intervention were less adherent than white participants at all time points despite an individualized and accessible intervention [57].

## 5. Conclusions

Implementing the diet and exercise interventions for older Hispanics with type 2 diabetes was feasible. There were slight improvements in both intervention groups compared to the control group. The diet-only group tended to have larger improvements on body composition measures (especially in muscle mass). The diet and exercise group tended to have larger improvements on physical function (especially in chair stands). There was a high rate of attrition (participants that quit the program), especially in the combined diet and exercise group, but those who completed the intervention tended to have improvements in body composition and physical function.

## Figures and Tables

**Table 1 ijerph-18-08019-t001:** Trial arms: duration, frequency, and details.

Arms	Interventions
Diet + exercise (*n* = 8)	2×/week, 30 min group exercise, 30 min of walking, & 2×/week, 30 min group nutrition sessions.
Diet-only (*n* = 6)	2×/week, 30 min group sessions.
Control (*n* = 15)	Completed all assessments but did not receive any intervention.

**Table 2 ijerph-18-08019-t002:** Body composition measures per assessment, per group.

Variables	Group	Baseline	Month 3	Time (F; *p*)	Interaction (F; *p*)
Weight (kg)	Diet + exercise	78 ± 17	78 ± 17	0.1; 0.737	0.2; 0.785
	Diet	73 ± 18	72 ± 19	0.1; 0.711	
	Control	72 ± 10	71 ± 9	0.5; 0.500	
Waist circumference (cm)	Diet + exercise	103 ± 16	108 ± 11	2.9; 0.099	2.3; 0.119
	Diet	106 ± 11	103 ± 17	0.9; 0.353	
	Control	100 ± 9	98 ± 9	0.9; 0.364	
Hip circumference (cm)	Diet + exercise	112 ± 11	114 ± 11	0.8; 0.387	0.9; 0.419
	Diet	106 ± 18	110 ± 13	1.6; 0.220	
	Control	105 ± 9	104 ± 9	0.1; 0.758	
Body Mass Index (km/m^2^)	Diet + exercise	32 ± 6	31 ± 6	0.1; 0.764	0.0; 0.991
	Diet	29 ± 7	29 ± 7	0.0; 0.888	
	Control	28 ± 4	28 ± 3	0.2; 0.636	
Waist-to-hip ratio (U)	Diet + exercise	0.93 ± 0.11	0.95 ± 0.05	0.6; 0.460	2.1; 0.138
	Diet	1.02 ± 0.10	0.94 ± 0.12	4.3; 0.048 *	
	Control	0.95 ± 0.07	0.94 ± 0.07	0.3; 0.566	
Body Fat (%)	Diet + exercise	39 ± 14	36 ± 14	1.1; 0.301	0.5; 0.588
	Diet	37 ± 16	34 ± 11	2.0; 0.168	
	Control	40 ± 10	39 ± 9	0.2; 0.697	
Muscle mass (%)	Diet + exercise	26 ± 7	27 ± 6	0.9; 0.349	1.7; 0.196
	Diet	26 ± 6	29 ± 5	7.5; 0.011 *	
	Control	25 ± 5	26 ± 4	0.8; 0.380	

Mean ± SD; * *p* < 0.05. ANOVA and Tukey’s multiple comparison test HSD.

**Table 3 ijerph-18-08019-t003:** Upper and lower body strength measures per assessment, per group.

Variables	Group	Baseline	Month 3	Time (F; *p*)	Interaction (F; *p*)
Grip strength (kg)	Diet + exercise	23 ± 7	22 ± 8	0.5; 0.474	0.782; 0.468
	Diet	19 ± 7	18 ± 5	1.4; 0.252	
	Control	21 ± 6	21 ± 8	0.2; 0.697	
Chair stands (rep)	Diet + exercise	9 ± 4	10 ± 5	5.6; 0.025 *	1.735; 0.196
	Diet	6 ± 6	6 ± 6	0.0; 1.000	
	Control	7 ± 4	7 ± 4	0.1; 0.703	

Mean ± SD; * *p* < 0.05. ANOVA and Tukey’s multiple comparison test HSD.

**Table 4 ijerph-18-08019-t004:** Diabetes biomarker and blood pressure per assessment, per group.

Variables	Group	Baseline	Month 3	Time (F; *p*)	Interaction (F; *p*)
HbA1c (%)	Diet + exercise	7 ± 1	7 ± 1	0.0; 0.918	0.072; 0.931
	Diet	7 ± 1	7 ± 0	0.0; 0.905	
	Control	7 ± 1	7 ± 1	0.1; 0.707	
Capillary glucose (g/dL)	Diet + exercise	134 ± 43	162 ± 89	2.8; 0.107	1.198; 0.318
	Diet	174 ± 60	169 ± 73	0.1; 0.811	
	Control	127 ± 31	125 ± 23	0.0; 0.905	
Systolic pressure (mm/Hg)	Diet + exercise	145 ± 13	136 ± 14	3.0; 0.096	1.042; 0.367
	Diet	129 ± 18	131 ± 13	0.1; 0.726	
	Control	134 ± 17	132 ± 15	0.3; 0.563	
Diastolic pressure (mm/Hg)	Diet + exercise	79 ± 7	78 ± 9	0.1; 0.750	0.364; 0.699
	Diet	73 ± 11	72 ± 13	0.1; 0.769	
	Control	79 ± 13	81 ± 9	0.6; 0.466	

HbA1c: glycated hemoglobin; mean ± SD; ANOVA and Tukey’s multiple comparison test HSD.

**Table 5 ijerph-18-08019-t005:** Observed trends in the variables studied.

Variables Studied	Group(s) That Presented Some Change
Weight (kg)	--
Waist circumference (cm)	--
Hip circumference (cm)	>Diet only (NS), >Diet + exercise (NS)
Waist-to-hip ratio (U)	Diet only
Body Mass Index (km/m^2^)	--
Body Fat (%)	<Diet only (NS), <Diet + exercise (NS)
Muscle mass (%)	Diet only
Grip strength (Kg)	--
Chair stands (rep)	Diet + exercise
Blood pressure (mmHg)	--
HbA1c (%)	--
Capillary glucose (g/dL)	<Diet only (NS), >Diet + exercise (NS)

NS = not statistically significant change.

## Data Availability

Data supporting reported results can be requested to the corresponding author.

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
