# Peer review of "Effects of Exercise and Diet on Body Composition and Physical Function in Older Hispanics with Type 2 Diabetes"

_ijerph, 2021, doi:10.3390/ijerph18158019_

Round 1

Reviewer 1 Report

The manuscript entitled "Effects of exercise and diet on body composition and physical function of older hispanics with type 2 diabetes" concerns a serious problem which is type 2 Diabetes mellitus. The manuscript is well written, however there are few points that should be revised.

Method section:

  • please describe with details the nutrition intervention. You have written " the interventions were delivered twice a week" but it is not clear if participants were on a diet all the time or just 2 days per week. Did they receive all products (2 times a week) that are needed to prepare all meals for the whole week or ready meals but for 2 days?  What kind of diet was that?
  • table 1: it would be good to add number of participants (for each group)

Results section:

  • number of Tables are wrong (in the text)

Author Response

Thank you!

The following explanation was included:

The educational nutrition intervention was based on Dietary Guidelines for Older Americans and Diabetes.org’s food and Nutrition Recommendations and tailored to the participants based on the food preferences, affordability and availability. The educational intervention was on healthy eating to improve diabetes management (e.g. glycemic control and diet quality). No meals were provided directly to the participants. The participants received evidenced-based diet education for older adults with type 2 diabetes; the following topics were presented interactively 2 times/week (30 min/session):

1.What’s in food for me? Senior nutrition and food safety;

2.Healthy foods/superfoods/healthy foods on a budget;

3.Meal plans, portion & serving sizes, and reading food labels;

4.Modifying recipes – healthy recipes your way;

5.Dietary supplements and tips for staying hydrated;

6.Closer look at reading food labels & meal planning (tips on eating out).

Also, we included the number of participants per group on Table 1, and corrected the number of the tables.

Thank you for helping us improve our manuscript.

Reviewer 2 Report

Congratulations to the authors of the work. I have enjoyed reading the manuscript.

I suggest adding a table with the corresponding explanatory text where the different antidiabetic treatments corresponding to each of the groups is reflected, due to the influence they may have on the study variables. 

Likewise, I encourage the authors to extend the study period when the circumstances of the pandemic allow it.

Author Response

Thank you! We included the following table with the expected and achieved changes per intervention group on the variables of the study:

Table 5.  Observed trends on the variables studied.

Variables studied

Group(s) that presented some change

Weight (kg)

--

Waist circumference (cm)

--

Hip circumference (cm)

>Diet only (NS), >Diet+exercise (NS)

Waist-to-hip ratio (U)

Diet only

Body Mass Index (km/m2)

--

Body Fat (%)

<Diet only (NS), <Diet+exercise (NS)

Muscle Mass (%)

Diet only

Grip strength (Kg)

--

Chair stands (rep)

Diet+exercise

Blood pressure (mmHg)

--

HbA1c (%)

--

Capillary glucose (g/dL)

<Diet only (NS), >Diet+exercise (NS)

Thank you for your encouragement. We hope we can continue. Our plan is to use our preliminary findings presented in this manuscript to apply for funding to do a larger trial as soon as the conditions permit. ?